# Novel Polymer Composites for Lead-Free Shielding Applications

**DOI:** 10.3390/polym16071020

**Published:** 2024-04-08

**Authors:** Mazen A. Baamer, Saad Alshahri, Ahmed A. Basfar, Mohammed Alsuhybani, Alhanouf Alrwais

**Affiliations:** 1M.Sc. in Nuclear Engineering Program, College of Engineering, King Saud University, Riyadh P.O. Box 145111, Saudi Arabia; abasfar@ksu.edu.sa; 2Engineering and Project Management Sector, King Abdullah City for Atomic and Renewable Energy (K.A. CARE), Riyadh 11451, Saudi Arabia; 3Nuclear Technologies Institute, King Abdulaziz City for Science and Technology (KACST), Riyadh 11442, Saudi Arabia; sshahri@kacst.gov.sa (S.A.); sohybani@kacst.edu.sa (M.A.); aalrwais@kacst.gov.sa (A.A.); 4Mechanical Engineering Department, College of Engineering, King Saud University, Riyadh 4545, Saudi Arabia

**Keywords:** polymeric micro-composite, polymeric nanocomposite, LDPE, radiation shielding, attenuation of X-ray radiation

## Abstract

Polymer nanocomposites have recently been introduced as lead-free shielding materials for use in medical and industrial applications. In this work, novel shielding materials were developed using low-density polyethylene (LDPE) mixed with four different filler materials. These four materials are cement, cement with iron oxide, cement with aluminum oxide, and cement with bismuth oxide. Different weight percentages were used including 5%, 15%, and 50% of the cement filler with LDPE. Furthermore, different weight percentages of different combinations of the filler materials were used including 2.5%, 7.5%, and 25% (i.e., cement and iron oxide, cement and aluminum oxide, cement and bismuth oxide) with LDPE. Bismuth oxide was a nanocomposite, and the remaining oxides were micro-composites. Characterization included structural properties, physical features, mechanical and thermal properties, and radiation shielding efficiency for the prepared composites. The results show that a clear improvement in the shielding efficiency was observed when the filler materials were added to the LDPE. The best result out of all these composites was obtained for the composites of bismuth oxide (25 wt.%) cement (25 wt.%) and LDPE (50 wt.%) which have the lowest measured mean free path (MFP) compared with pure LDPE. The comparison shows that the average MFP obtained from the experiments for all the eight energies used in this work was six times lower than the one for pure LDPE, reaching up to twelve times lower for 60 keV energy. The best result among all developed composites was observed for the ones with bismuth oxide at the highest weight percent 25%, which can block up to 78% of an X-ray.

## 1. Introduction

In the pursuit of environmentally sustainable alternatives for effective radiation shielding, this study delves into the potential of low-density polyethylene (LDPE) and LDPE-based composites. Recognizing the inherent environmental and health concerns that are tied to lead-based shielding, the primary focus of this work revolves around the development and characterization of lead-free polymer composites. We specifically explore the integration of LDPE in various formulations, leveraging its known attributes of flexibility, durability, and ease of processing. Studies have shown the development of lead-free polymers which demonstrate improved photon attenuation radiation shielding across various industries compared to traditional lead-based shields which pose toxicity and handling challenges [1,2,3,4].

LDPE stands out as a compelling candidate for radiation shielding applications due to its inherent properties. This study not only investigates the shielding capabilities of pure LDPE but we also extended our exploration to composite materials by introducing cement into LDPE matrices. The inclusion of cement is aimed at enhancing the material’s radiation attenuation properties, introducing a novel dimension to the lead-free polymer composite design. Polymer composites, particularly those reinforced with micro- or nano-scale fillers, exhibit improved radiation shielding capabilities. Polymeric composites featuring reinforcement with metal and metal oxides micro and nanoparticles stand as the prevailing choice for shielding against X- and γ-rays within scientific literature, owing to their demonstrated efficacy in radiation attenuation. These advancements suggest a practical alternative to traditional materials like lead, with enhanced efficiency in stopping radiation [3,4,5,6].

Nambiar et al. presented innovative lead-free shielding solution—polydimethylsiloxane (PDMS) nanocomposites with 44.44% bismuth oxide. Proven to be effective against diagnostic X-rays, the PDMS material matched the attenuation of a 0.25 mm thick pure lead sheet, offering a lighter, easily moldable, and paintable alternative [7].

Alavian investigated the efficacy of LDPE/metal oxide composites in gamma ray shielding. Notably, a blend of 50% PbO and WO_3_ outperformed ZnO and TiO_2_, showcasing superior attenuation. By employing the Monte Carlo simulation, the investigation revealed that lowering the incident photon energy improved the composite’s performance, highlighting the potential of using nanoparticles for effective radiation shielding, particularly in scenarios involving low photon energies [8].

Almurayshid conducted a study on using new lead-free composite materials for radiation shields. The aim was to create cost-effective, lightweight, and non-toxic polymer-based shields. Seven materials were examined, with EVA + SiC (30%) and EVA + Si (15%) + B4C (15%) outperforming pure EVA, requiring lesser thicknesses to stop 50% of incident photons. EVA + SiC (30%) effectively blocked 90–91% of X-rays at around 80 keV, demonstrating promising results for enhanced shielding performance [9].

Additional work was conducted by Alshahri on the preparation, characterization, and application in X-ray shielding of a LDPE/Bismuth oxide nanocomposite [10]. This study developed a new nanocomposite as a shield to block the X-ray rather than using lead. The developed material was based on an LDPE polymer and bismuth oxide (Bi_2_O_3_) with three different weight percentages of Bi_2_O_3_: 5%, 10%, and 15%. The study investigated the developed nanocomposites with regard to its thermal, structural, mechanical properties, physical features, and radiation shielding efficiency. The results were optimistic for radiation shielding as well as for improving the composite’s density while increasing its tensile strength and tensile modulus. The highest efficiency was achieved when the nanofiller added into the LDPE was 15% Bi_2_O_3_. This composite required the lowest thickness to attenuate 50% of the X-rays. Moreover, 80% of the X-rays at 47.9 keV can be blocked using LDPE + 15% Bi_2_O_3_. It is recommended to have higher percentages of the filler material as well as to increase the shielding thickness in order to reach a high level of radiation blockage.

Our research design encompasses six distinct formulations: pure LDPE, LDPE with cement, LDPE with cement and iron powder, LDPE with cement and aluminum oxide, LDPE with bismuth oxide, and LDPE with cement and bismuth oxide. Our approach involves a systematic synthesis process, ranging from the precise preparation of materials to rigorous testing protocols. The overarching objective is to comprehensively assess the radiation attenuation efficacy of these materials, spanning their initial formulation and throughout the entirety of testing. This comprehensive examination seeks to yield valuable insights into the suitability of LDPE and LDPE-based composites as robust and effective lead-free alternatives for radiation shielding [3,5,6,10,11].

The objectives of this study involve the synthesis and characterization of these lead-free polymer composites, the evaluation of their radiation attenuation capabilities, and a comparative analysis of their performance. The anticipated findings are poised to significantly contribute to ongoing efforts in developing environmentally friendly and health-conscious solutions for radiation shielding applications. As industries and research communities increasingly prioritize sustainable technologies, the outcomes of this investigation hold the potential to drive practical advancements toward safer and more eco-friendly practices within ionizing radiation protection [4,10].

## 2. Materials and Methods

To comprehensively evaluate the lead-free polymer composites developed in this study, a multi-faceted approach employing a range of analytical techniques is essential. In addition to assessing radiation shielding efficacy, this research integrates thermal gravimetric analysis (TGA), differential thermogravimetric analysis (DTGA), tensile testing for mechanical strength, elasticity, and deformation characteristics, density measurement for insights into compactness and uniformity, Fourier transform infrared spectroscopy (FTIR), scanning electron microscopy (SEM), and energy dispersive X-ray spectroscopy (EDS). This comprehensive suite of analyses aims to provide a thorough understanding of the physical, thermal, and chemical properties of the developed lead-free polymer composites [12,13,14].

### 2.1. Materials

In pursuit of optimal shielding capabilities, LDPE (LDPE HP 0722N, SABIC, Riyadh, Saudi Arabia) was deliberately chosen as the base polymer due to its inherent attributes of flexibility, durability, and processability. To optimize the shielding efficacy against X-rays, various filler materials were introduced into the LDPE matrix. These fillers, carefully selected for their potential to enhance radiation attenuation, contribute to the overall effectiveness of the polymer composites in shielding applications.

In our study, various filler materials were strategically incorporated into the LDPE matrix to augment its shielding capabilities. Cement, sourced from Ordinary Portland Cement (OPC) provided by YAMAMA Cement Co., Riyadh, Saudi Arabia, was chosen due to its high density and frequent utilization in nuclear power plant construction, particularly in containment building structures. Additionally, its cost-effectiveness made it a practical choice for our research.

Fe_2_O_3_ powder, obtained from Sigma Aldrich, St. Louis, MO, USA, was selected for its high density, which contributes to effective radiation attenuation. Similarly, Al_2_O_3_ powder from Sigma Aldrich, St. Louis, MO, USA, was chosen for its density and its common use in enhancing cable properties, making it a suitable candidate for our study.

Furthermore, nano-sized Bi_2_O_3_ particles, also sourced from Sigma Aldrich, St. Louis, MO, USA, were selected due to their high density and promising results being reported in existing literature. These particles offer potential advantages in radiation shielding due to their size and composition.

The selection of these filler materials was based on their respective densities and demonstrated effectiveness in enhancing radiation shielding properties, aligning with the objectives of our study. The investigated fillers encompassed the following formulations seen in Table 1.

The materials investigated in this study offer a unique advantage when applied for shielding purposes, presenting a novel approach to enhance protective measures in various applications. This innovative utilization highlights their potential for advancing the field of lead-free X-ray shielding, addressing critical needs for safer and more sustainable alternatives in radiation protection.

### 2.2. Methodology

The methodology steps are shown below starting from weighting the materials to shaping them into the required shapes:

Weighing the materials: Utilize an electrical balance (Sartorius Analytical, Karlsruhe, Germany) with an accuracy of ±0.0001 g to weigh the samples.Mixing Process:^1.^ Use an internal mixer (Internal mixer 350 S, Brabender, Duisburg, Germany) set at 175 °C.^2.^ Begin mixing LDPE alone.^3.^ Gradually increase the rotation speed from 40 RPM to achieve the best mixing results, increasing in 5 RPM increments.^4.^ Introduce fillers gradually after reaching maximum speed.^5.^ Continue mixing for 10 min.Rolling the Mixture:^6.^ Employ a two-roll mill (Prep-Mill, Brabender, Duisburg, Germany) set at 170 °C and 30 RPM.^7.^ Roll the mixture for 2–5 min.^8.^ Allow the mixture to cool.Shaping and Pressing:^9.^ Cut the mixture into pieces in preparation for the pressing machine (P400 PM, Collin, Germany).^10.^ Use a stainless-steel frame measuring 100 × 100 × 2 mm^3^ to shape the sample.^11.^ Press the sample at 170 °C, starting at 30 kN and gradually increasing up to 120 kN over 10 min.^12.^ Let the sample cool at room temperature.Preparing Testing Samples:^13.^ Cut the sample into required shapes for testing, such as discs for shielding and dumbbell shapes for tensile and other tests.

## 3. Characterization

This section is dedicated to thoroughly analyzing the materials developed in this study, specifically focusing on their potential as radiation shields. The following points outline the specific tests employed in this study.

### 3.1. X-ray Test

This study focuses on a comprehensive assessment of the samples as potential shielding materials through X-ray testing. To address the considerable sample volume, a strategic focus was applied to three concentrations—low, medium, and high—for each sample. Precision in characterization was heightened by employing digital measurements to determine and average the thickness of each sample, considering data from five discs. This methodical approach ensures not only accuracy but also efficiency in the evaluation process, providing an in-depth perspective on the effectiveness of the samples in attenuating X-rays.

The linear attenuation coefficient (*μ*) and the mass attenuation coefficient (*μ_m_*) were calculated using the following equations, Lambert–Beer Law:(1)$$\frac{I}{I_{0}}=e^{-\mu x}$$
(2)$$\frac{I}{I_{0}}=e^{-(\frac{\mu }{\rho })\rho x}$$
(3)$$\mu_{m}=\frac{\mu }{\rho }$$
where $I_{0}$ is the initial intensity of the radiation, I is the intensity of the radiation after passing the shielding material with thickness (*x*). ρ is the material composite density. The configuration of the experiment set up is shown in Figure 1. Eight X-ray energies were used in the study, as listed in Table 2. The sample was shaped into a disc with a diameter of 2 cm, and it was then fixed in the space between the sample holder to measure and then calculate the attenuation. The irradiation procedures were conducted using a 1.80 cm^2^ sized collimator and a 30-s-long acquisition time. The μ values for the composite samples were determined using the intensity values with no disks and with discs for I_o_ and I, respectively. Table 2 shows the X-ray irradiation qualities used using a narrow beam spectrum condition allowing only primary photons to pass through the attenuating material to contribute to the detected signal. Using SpekCalc software, and for illustration purposes, Figure 2 confirms the effective energy stated in ISO-4037 [15] for the applied voltage of 120 kVp and with the filters provided in Table 2. The X-ray intensity decreased as the energy decreased until it reached the threshold energy, which, in this case, was 47 KeV due to the filter material and thickness used.

In this study, the half value layer (HVL) and mean free path (MFP) were measured. These values are mainly dependent on the energy and the thickness. The amount of radiation that would be 50% is the HVL which is the shielding thickness that reduces the radiation by half. The MFP is the material thickness required to reduce the radiation to 36.8%. The radiation shielding efficiency (RPE) is the intensity of the radiation values measured before the shielding sample and after we added the shielding sample to indicate the ability of the shielding composite samples to block the radiation. The equation for this is shown below.
(4)$$HVL=\frac{0.693}{\mu }$$
(5)$$MFP=\frac{1}{\mu }$$
(6)$$RPE=\left(1-\frac{I}{I_{0}}\right)\times 100$$

### 3.2. Thermal Analysis by TGA and DTG

The thermal behavior of the samples was evaluated using a thermogravimetric analyzer and differential thermogravimetric analysis (TGA and DTG, Model TGA 1 from Perkin Elmer, Shelton, CT, USA). Each sample was heated from room temperature to 700 °C at a rate of 10 °C min^−1^.

### 3.3. Tensile Testing

Tensile testing was performed using a tensile machine (Instron 5982, Grove City, PA, USA) in accordance with ASTM D638 standards. Dumbbell samples, cut from pressed sheets with a thickness of 2 mm, were tested at a crosshead speed of 50 mm/min. The results represent the average of five measurements. The tensile strength, Young’s modulus, and elongation at break of the composites were calculated.

### 3.4. Density Measurement

The experimental densities of the mixtures were measured using the Archimedes method (with ethanol as an immersing medium). The densities of all of the developed samples were calculated using the Mettler Toledo XS204 instrument (Greifensee, Switzerland) as shown in the equation below.
(7)$$\rho =\frac{w_{air}}{\left(w_{air}-w_{ethanol}\right)}\times {(\rho }_{ethanol}-\rho_{air})+\rho_{air}$$
where
$$\rho :sample\;density\;\left(\frac{g}{cm^{3}}\right).$$
$$w_{air}:sample\;weight\;in\;air\;(g).$$
$$w_{ethanol}:sample\;weight\;in\;ethanol\;(g).$$
$$\rho_{etanol}:density\;of\;the\;etanol\left(\frac{g}{cm^{3}}\right).$$
$$\rho_{air}:density\;of\;the\;air\left(\frac{g}{cm^{3}}\right).$$

### 3.5. Scanning Electron Microscopy (SEM) and Energy Dispersive X-ray Spectroscopy (EDS)

In this study, SEM was used to investigate the microstructure of the eight highest-performing samples selected from the prepared set. These samples, chosen for their exceptional X-ray shielding properties, were scrutinized using SEM to gain insights into the structural features responsible for their superior performance. EDS was applied to the top-performing eight samples selected from the prepared set. These samples, recognized for their outstanding X-ray shielding properties, were examined using EDS to uncover the elemental composition and distribution contributing to their superior performance.

#### Analysis by FTIR Spectroscopy

FTIR measurements were conducted in this study to evaluate the molecular composition of the prepared samples (Spectrum one FT-IR Spectrometer, Perkin Elmer, Shelton, CT, USA). Small pieces were taken from each sample, and a calibration process was employed before each test to ensure accurate results. This meticulous approach enhanced the reliability of the FTIR data, facilitating a comprehensive analysis of the samples’ molecular characteristics.

## 4. Results and Discussion

In the present study, we showcase the outcomes of various tests, including tensile testing, thermal analysis (TGA and DTG), SEM and EDS, FTIR, and X-ray assessment. These tests collectively validate the materials’ capabilities, offering valuable insights into their suitability as shielding materials. For the X-ray test, results for all the developed composites will be evaluated in comparison to pure LDPE. In contrast, for the other tests, the most promising results from each composite will be evaluated focusing on those that offer the best X-ray shielding properties. This selection includes eight composites, which have the highest filler material concentrations. In the case of the bismuth composite, we selected all of them [16,17,18].

### 4.1. Tensile Test

As listed in Table 3 and Figure 3, our evaluation of the samples revealed variations in tensile strength and elongation at break relative (normalized) to pure LDPE. L50C50 demonstrated the highest tensile strength at 108.69% of the pure LDPE, indicating superior mechanical strength with a limited elongation at break of 3.24%. L50C25I25, in contrast, exhibited a lower tensile strength at 63.33% and a relatively small elongation at break of 2.25%. L50C25Alo25 showcased a robust tensile strength at 87.80%, while its elongation at break was notably low at 1.34%, indicating a preference for applications prioritizing strength over ductility. In contrast, L90Bio10 combined a high tensile strength of 93.83% with outstanding ductility, as indicated by an elongation at break of 95.02%, making it an excellent candidate for applications requiring both strength and deformation tolerance. L95C2.5Bio2.5 offered a balanced combination of tensile strength (81.83%) and elongation at break (16.41%), making it versatile for various tensile testing scenarios. L85C7.5Bio7.5 demonstrated a good balance between tensile strength (88.20%) and elongation at break (10.12%). L50C25Bio25 shared the same tensile strength as L85C7.5Bio7.5 but exhibited a lower elongation at break of 4.16%. These results underscore the significance of understanding both tensile strength and elongation at break properties in the context of tensile testing, and each sample’s characteristics provide valuable insights into its material behavior under uniaxial tensile loading conditions [16].

### 4.2. Thermal Analysis by TGA and DTGA

In our research, we thoroughly investigated how the materials responded to temperature, heating them from 25 °C to 800 °C at a steady 10 °C/min. Figure 4 and Figure 5 show the thermal behavior of the most promising samples relative to pure LDPE. Notably, all of the composites showed stability up to 450 °C, displaying minimal mass loss in the TGA thermograms. At 550 °C, LDPE underwent a substantial weight loss leaving only 1.71% of its original mass. Meanwhile, the L50C25Bio25 composite exhibited a significant performance in terms of remaining weight, reaching 47.1%. This improvement in thermal stability is attributed to the inclusion of cement and a higher weight percentage of Bi_2_O_3_ compared to a previous study, where the composite showed only 13.2% of remaining weight at the same temperature [19]. Similarly, the LDPE with cement at 50 wt.% and LDPE with cement and Fe_2_O_3_ at 25 wt.% each exhibited weight losses of 48.3% and 49.4%, respectively, with the latter representing the highest observed value [14]. This is a clear confirmation of the fact that the thermal stability of various inorganic compounds is greater than those of pure polymer. Furthermore, the overall increase in thermal stability of such composites could be linked to an increase in the density of the composite. The observed superiority in thermal stability of the inorganic compounds compared to pure polymer is evident. The incorporation of inorganic materials, like Bi_2_O_3_ and Fe_2_O_3_, strengthens the composite’s structure against thermal degradation, making these materials promising for applications requiring enhanced thermal performance. The introduction of an inorganic filler results in a significant increase in the overall thermal stability of the polymers with all of the filler materials introduced [20,21,22]. DTG thermograms, delineating the rate of material loss over time, showcase distinctive characteristics. Specifically, for pure LDPE, the lowest peak is observed at 516.29 °C, indicating a notably accelerated decomposition with a rate of −28.250%/min compared to the other composite materials. Regarding the composite with the highest concentration of Bi_2_O_3_, LDPE, and cement, the DTG thermogram displays a distinctive peak at 505.5 °C, suggesting a more gradual decomposition process with a rate of −13.0%/min. The composite with the highest concentration of Fe_2_O_3_, cement, and LDPE exhibits the lowest degradation rate at 513.8 °C with a rate of −12.49%/min. These results suggest a comparatively slower decomposition process for these composites, underscoring their enhanced thermal stability and resistance to degradation at higher temperatures.

### 4.3. SEM and EDS Analysis

Developed composites’ surfaces were examined using SEM in this study, with a specific focus on the composite with the most promising results in the X-ray properties test: the highest concentration of Bi_2_O_3_ with LDPE and cement. Figure 6 shows micrographs of both pure LDPE and the composite material at ×500 and ×3000 magnifications for each sample. The micrograph of pure LDPE illustrates a smooth surface without particles. In contrast, the composite with Bi_2_O_3_ clearly exhibits the presence of particles on its surface. This visual evidence supports the conclusion that the composite material incorporates discernible particles. The quality of the dispersion of the nanoparticles of Bi_2_O_3_ within the cement and LDPE was clearly observed. It is conceivable that certain particles may have settled during the mixture settling due to differences in chemical structure and physical properties between the polymer and other filler materials such as the nanoparticles of Bi_2_O_3_ and cement, despite uniform mixing. This behavior may be linked to its higher density and/or a lack of interaction or bonding with different polymer pellets during the heating process. The results of EDS are shown in Figure 7 and Figure 8 to further support the SEM micrographs of the developed composites [17].

### 4.4. Fourier Transform Infrared Spectroscopy (FTIR)

The FTIR analysis is shown in Figure 9. All of the spectra are similar to that of the pure LDPE. The same peaks were found for all samples in the range of 2850–2930 cm^−1^. This region represents the strong and sharp C-H stretching vibrations that are present in the samples containing the LDPE. Furthermore, a clear peak was shown at 1465 cm^−1^ for all of the tested samples. Moreover, a single vibration peak at 720 cm^−1^ was also found in all the samples [18].

### 4.5. X-ray Shielding Properties

#### 4.5.1. LDPE with Cement

Figure 10 shows the X-ray shielding properties of the developed composites compared to pure LDPE. Each figure shows four different curves including the pure LDPE and three different wt.% of the filler materials which are the cement starting with 5%, 15%, and 50% wt.%. As the concentration of the filler material increases the attenuation values (linear and mass) increase due to the density of the composite which mainly drives the values to be higher. However, it is the opposite way for the HVL and the MFP, so that the best material has a lower HVL and MFP. Finally, the average RPE for all of the investigated energy ranges of the materials made of LDPE with cement at 5%, 15%, and 50 are 14%, 12%, and 18%, respectively. The highest RPE was observed in the low energy range, which was 35% for the 50% concentration of cement. The results indicate that the average RPE for pure LDPE was nearly 10%, which is lower than the lowest percentage of the cement filler. Interestingly, at the lowest energy range, LDPE demonstrates its highest RPE, reaching 15%. These findings suggest that the presence of a cement filler enhances its efficiency, particularly at lower energy levels.

#### 4.5.2. LDPE with Cement and Fe_2_O_3_

Figure 11 shows the X-ray shielding properties of cement with Fe_2_O_3_ as a filler compared to LDPE. Each figure shows four different curves including the pure LDPE and three different wt.% of the filler materials which are cement and Fe_2_O_3_ with 2.5%, 7.5%, and 25% wt.% of each. As the concentration of the filler increases the attenuation values (linear and mass) increase due to the density of the composite which mainly drives the values to be higher. This is the opposite way for the HVL and the MFP where the best material has a lower HVL and MFP. Finally, the average RPEs for all of the investigated energy ranges of the materials made of LDPE with 2.5%, 7.5%, and 25% of cement and Fe_2_O_3_ were 11%, 12%, and 18%, respectively. The highest RPE was observed in the low energy range, which was 41% for the 25% concentration of cement and 25% of Fe_2_O_3_. The results reveal that the incorporation of two different filler materials leads to an increased RPE, as observed in LDPE with cement, where the highest RPE reached 35%. However, in this particular case, the introduction of the specified fillers yields an even higher RPE, reaching 41%. This suggests that the combination of these specific filler materials with LDPE enhances its radiation shielding properties further.

#### 4.5.3. LDPE with Cement and Al_2_O_3_

Figure 12 shows the X-ray shielding values of cement with Al_2_O_3_ as a filler material compared to LDPE. Each figure shows four different curves which are the pure LDPE and three different wt.% of the filler materials which are cement and Al_2_O_3_ starting with 2.5%, 7.5%, and 25% wt.% of each. As the concentration of the filler material increases the attenuation values (linear and mass) increase due to the density of the composite which mainly drives the values to be higher in this case only for the lower energy range. This was the opposite for the HVL and the MFP where the best material had a lower HVL and MFP, which were also in the lower energy range in this case. Finally, the average RPEs for all investigated energy ranges of the materials made of LDPE, 2.5%, 7.5%, and 25% cement and Al_2_O_3_ were 11%, 12%, and 13%, respectively. The highest RPE was observed in the low energy range, which was 25% for the 25% concentration of cement and 25% of Al_2_O_3_. The findings indicate that the inclusion of the Al_2_O_3_ filler material does not yield significant improvements in shielding properties, even with an increase in wt.%. Moreover, the comparison between LDPE with cement and LDPE with cement and Al_2_O_3_ composites reveals that LDPE with cement consistently exhibits better radiation properties. This suggests that the cement filler is more effective in enhancing the radiation shielding capabilities of LDPE compared to Al_2_O_3_.

#### 4.5.4. Bismuth Composite

Experiments were conducted on four composites with bismuth as a shielding material. The compositions of these composites are listed in Table 4.

The results are shown in Figure 13. Notably, the LDPE 50% with Bi_2_O_3_ 25% and cement 25% composite consistently demonstrated the most favorable outcomes across various parameters. This composite exhibited the highest X-ray attenuation values among all of the composites, making it the most effective in reducing the penetration of radiation. The LDPE 50% with Bi_2_O_3_ 25% and cement 25% composite recorded the lowest HVL, signifying its superior ability to reduce the intensity of radiation by half. Similarly, this composite had the smallest MFP, indicating that the radiation had the shortest distance to travel through the material. In addition, this composite demonstrated the highest RPE, particularly in the low-energy range, with an impressive 79%. This signifies its remarkable capability to shield against radiation effectively.

#### 4.5.5. High Shielding Efficiency Composite

Figure 14, Figure 15, Figure 16, Figure 17 and Figure 18 show the optimal MAC and LAC for all of the developed composites in comparison to pure LDPE. The findings conclusively demonstrate the exceptional performance of the bismuth composites across all of the X-ray tests. The Bi_2_O_3_ composite, particularly in high concentration, distinctly showcases superior values for RPE, the linear attenuation coefficient, and the mass attenuation coefficient compared to the other composites. Furthermore, the Bi_2_O_3_ composite at a high concentration displays the lowest values for HVL and MFP, establishing it as a highly promising material for efficient radiation shielding. The investigations also demonstrate that, among the discussed composites, the nanocomposite with Bi_2_O_3_ emerges as the most effective filler material in enhancing radiation shielding properties. This suggests that Bi_2_O_3_ nanocomposites outperform the other micro-composites considered in the study. The superior performance of the Bi_2_O_3_ nanocomposites highlights their potential for use in radiation protection applications. Nevertheless, the study also reveals promising results regarding the combination of nanocomposites with micro-composites. The mixing of nanocomposite materials with micro-composites demonstrates favorable outcomes in terms of radiation shielding properties. This finding suggests that synergistic effects may occur when these materials are combined, leading to an enhanced overall performance in shielding radiation.

## 5. Conclusions

The aim of this study was to develop potential lead-free and light-weight shielding materials based on a polymer composite using LDPE. The analysis of various tests for all of the prepared samples lead to promising materials for X-ray shielding, especially in the low energy range. The SEM micrographs showed that the interaction between the polymer and the fillers demonstrated a homogenous distribution and dispersion of various fillers. The average degradation temperature for all of the samples was 486 °C, showing thermal stability until 600 °C. A clear enhancement of the attenuation ability was observed when the percentage of filler was increased compared to the polymer. The best results were achieved for the composite of L50C25Bio25 (LDPE 50% with Bi_2_O_3_ 25% and cement 25% composite) with an ability to shield against X-rays of more than 78%. In summary, our study reveals the potential for innovative radiation protection solutions. Further research is essential to optimize the material’s performance in various shielding applications, including neutron shielding, increased thickness, and high-energy X-rays.

## Figures and Tables

**Figure 1 polymers-16-01020-f001:**
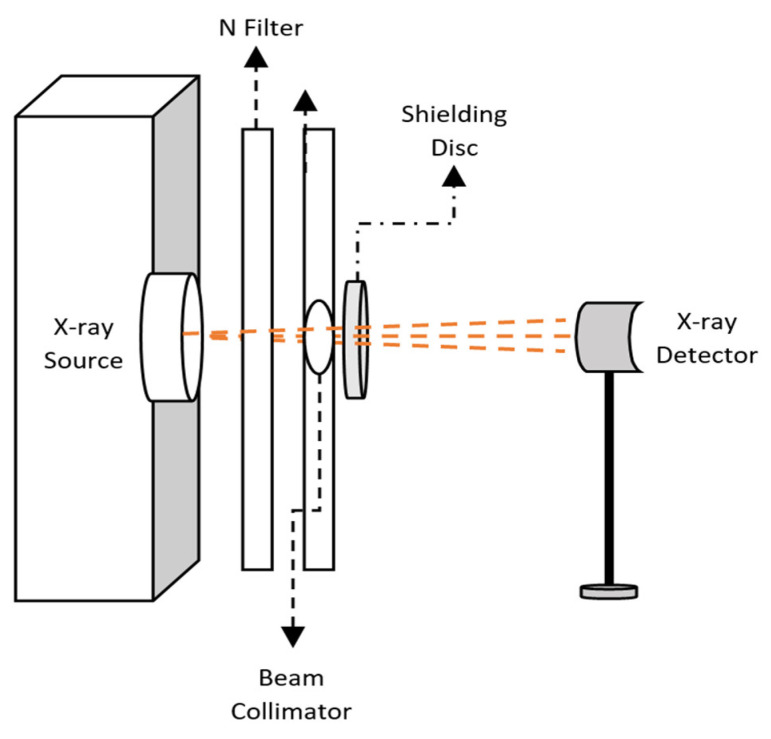
Experimental set-up for radiation attenuation experiments.

**Figure 2 polymers-16-01020-f002:**
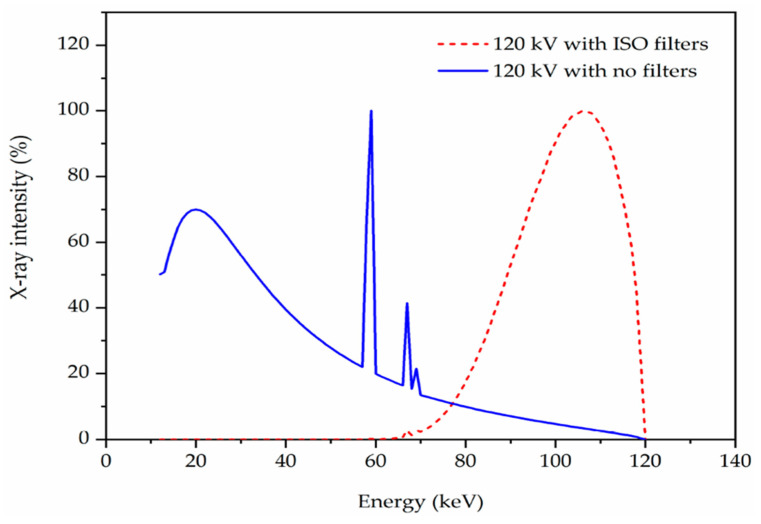
Distribution of the number of X-rays for 120 kVp extracted from SpekCalc with no filters and with ISO filter.

**Figure 3 polymers-16-01020-f003:**
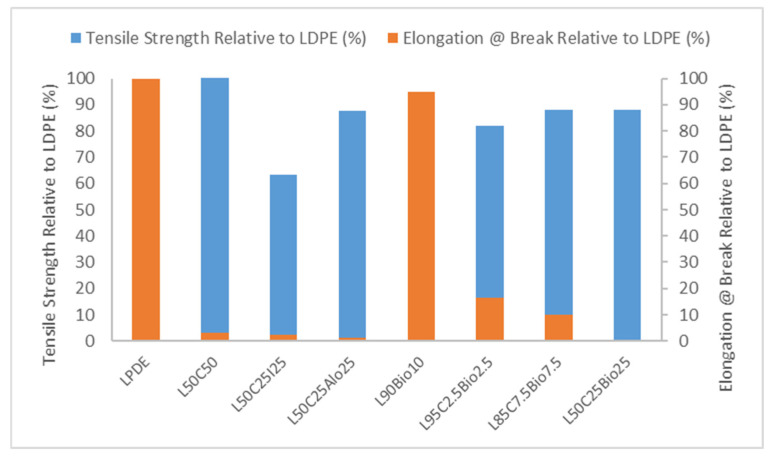
Tensile strength and elongation at break for the most promising results for X-ray shielding.

**Figure 4 polymers-16-01020-f004:**
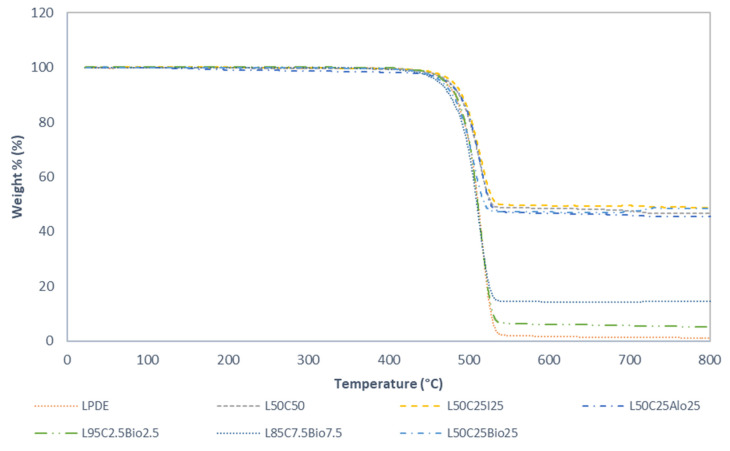
TGA thermograms of the most promising samples relative to pure LDPE.

**Figure 5 polymers-16-01020-f005:**
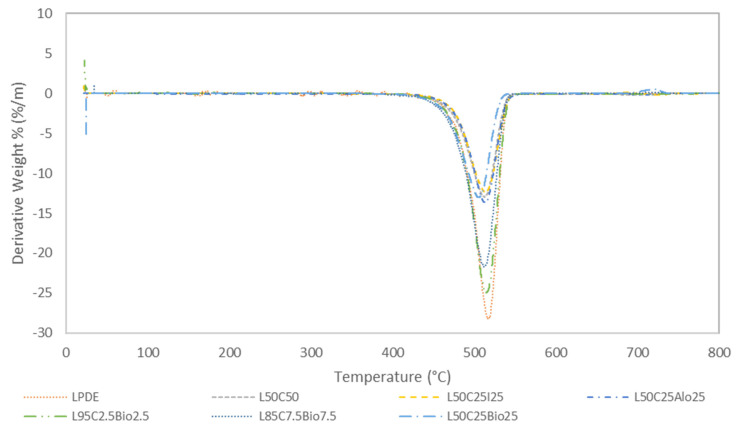
DTG thermograms of the most promising samples relative to pure LDPE.

**Figure 6 polymers-16-01020-f006:**
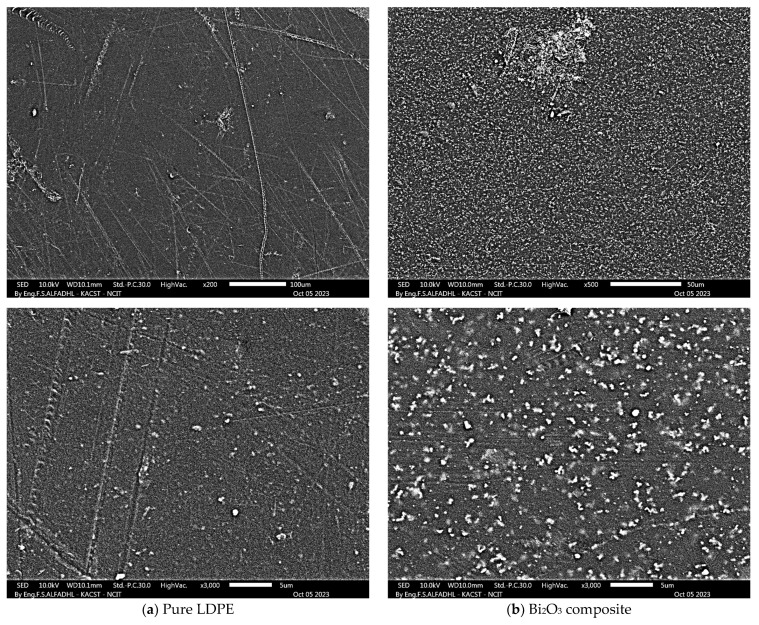
SEM micrographs for pure LDPE and LDPE 50 wt.% + Bi_2_O_3_ 25 wt.% + cement 25 wt.% composite.

**Figure 7 polymers-16-01020-f007:**
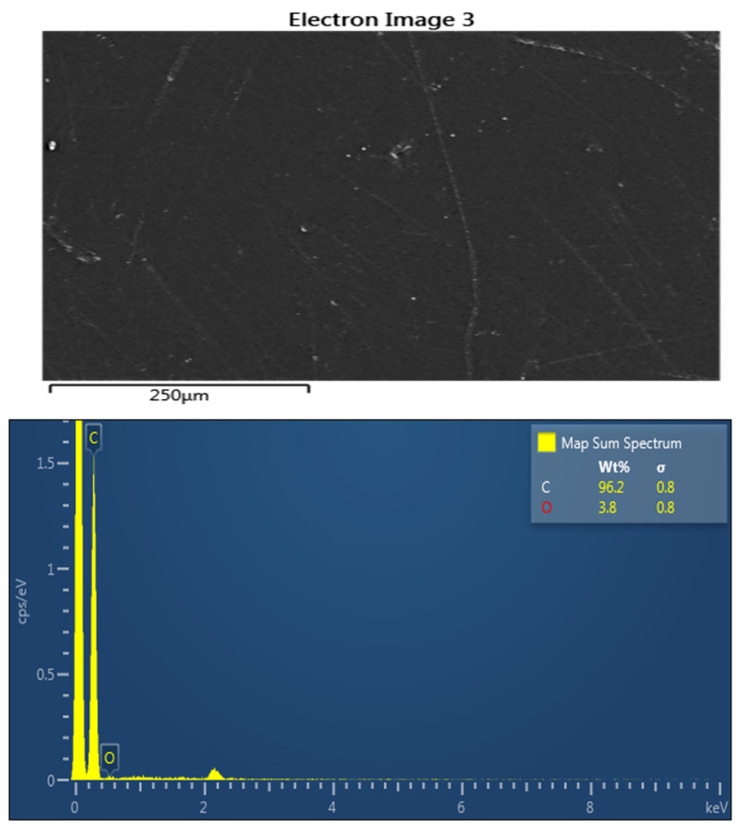
EDS spectrum for pure LDPE.

**Figure 8 polymers-16-01020-f008:**
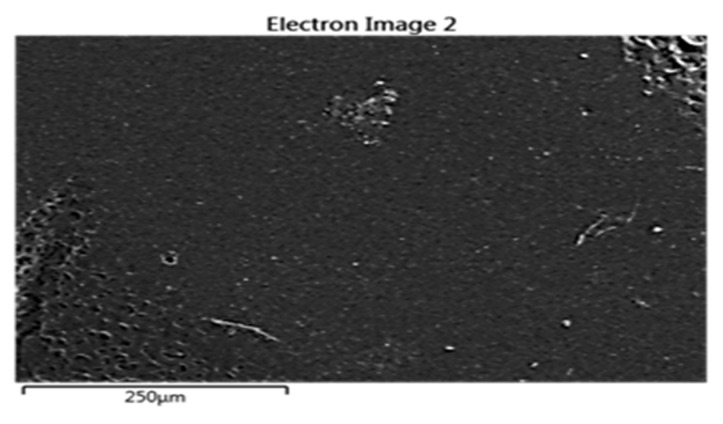
EDS spectrum for L50C25Bio25 composite.

**Figure 9 polymers-16-01020-f009:**
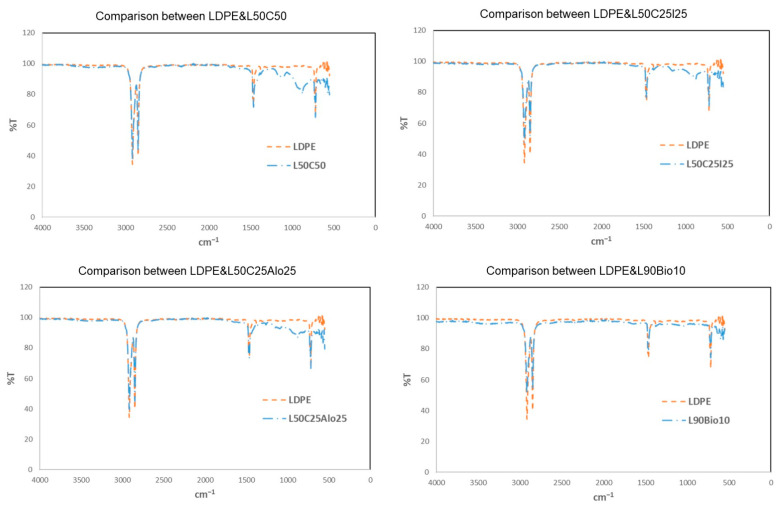
FTIR spectra for the most promising samples compared to pure LDPE.

**Figure 10 polymers-16-01020-f010:**
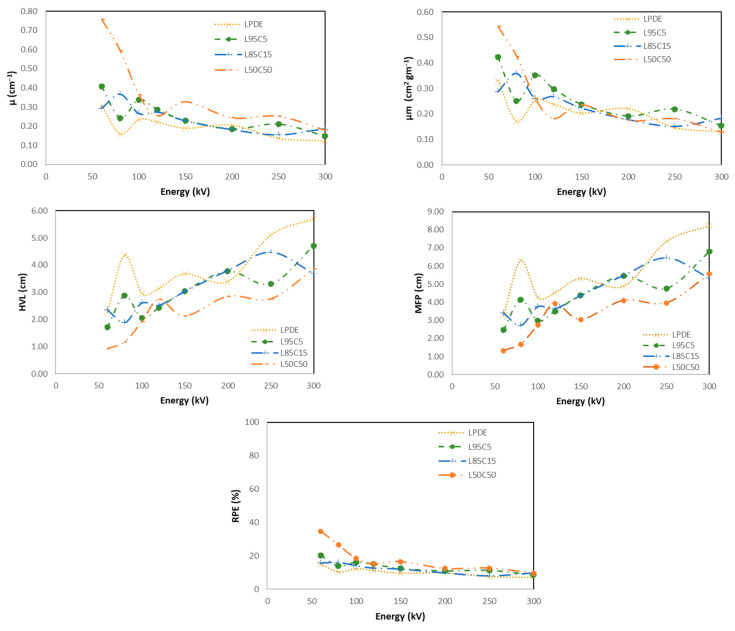
X-ray shielding properties of all developed composites compared to pure LDPE.

**Figure 11 polymers-16-01020-f011:**
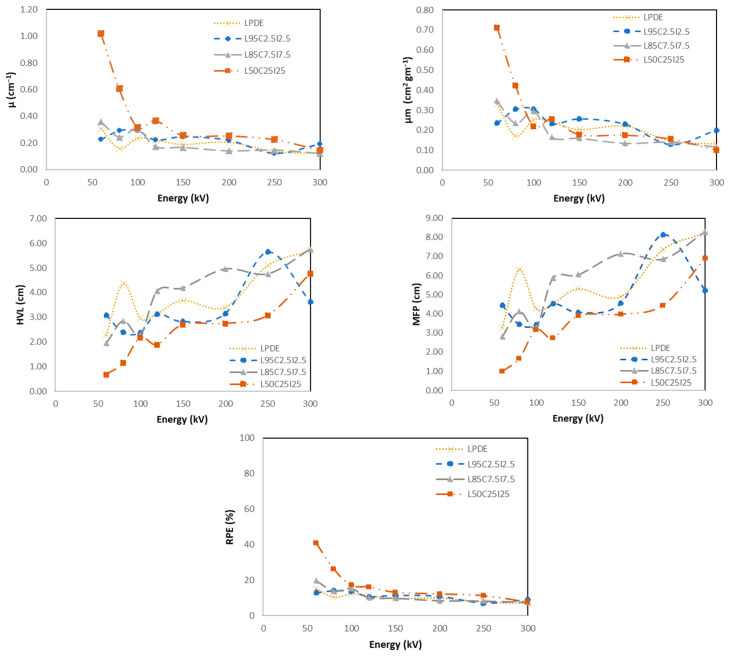
X-ray shielding properties of pure LDPE and composite of cement and Fe_2_O_3._

**Figure 12 polymers-16-01020-f012:**
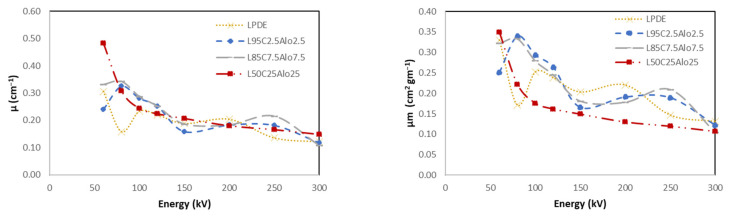
X-ray shielding properties of pure LDPE and composites of cement and AL_2_O_3._

**Figure 13 polymers-16-01020-f013:**
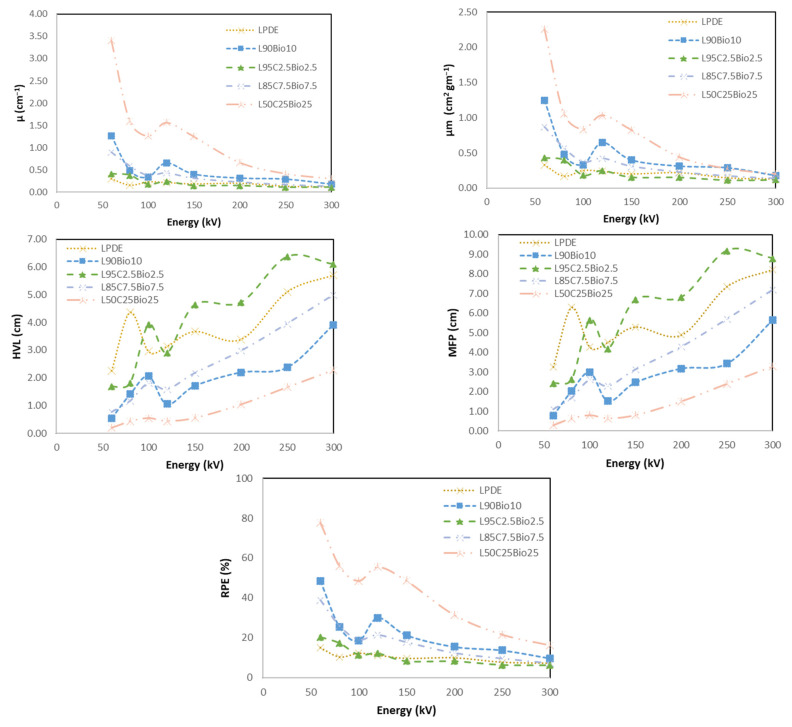
X-ray shielding properties of LDPE 50% with Bi_2_O_3_ 25% and cement 25% composites compared to pure LDPE.

**Figure 14 polymers-16-01020-f014:**
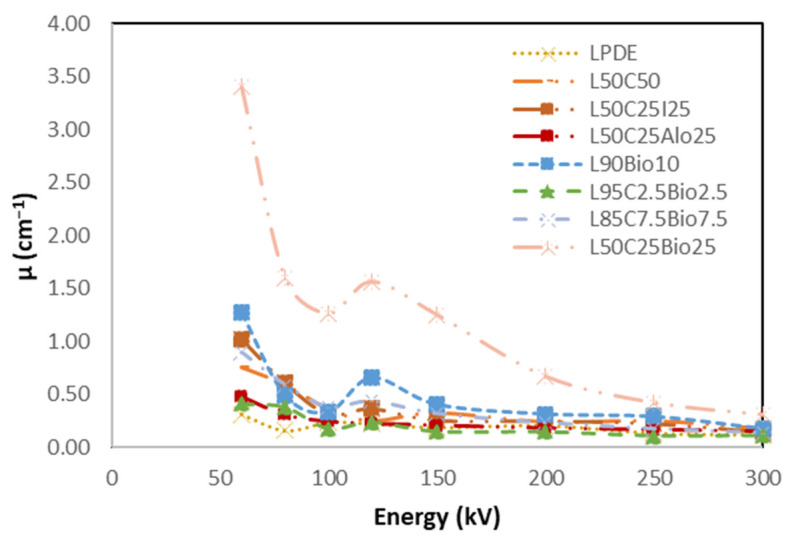
Comparison of MAC of the X-ray properties for the various prepared composites compared to pure LDPE.

**Figure 15 polymers-16-01020-f015:**
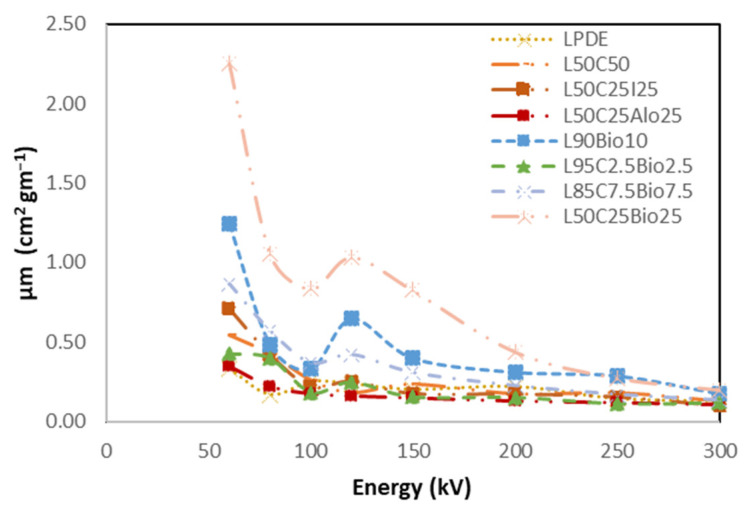
Comparison of LAC of the X-ray properties for all developed composites compared to pure LDPE.

**Figure 16 polymers-16-01020-f016:**
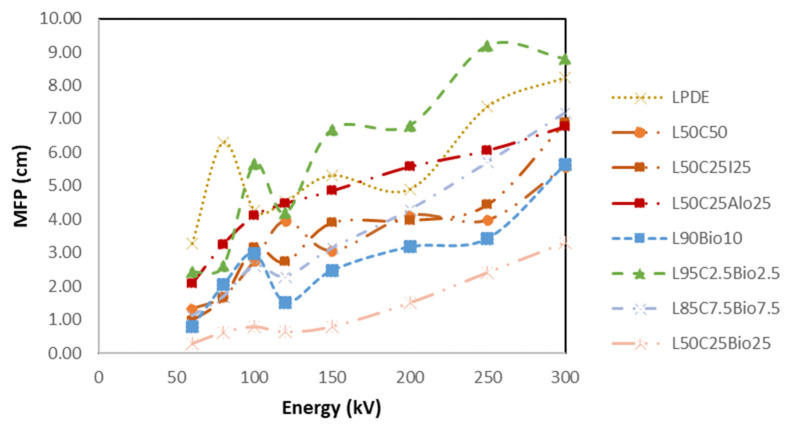
Comparison of MFP of the X-ray properties for the developed composites compared to LDPE.

**Figure 17 polymers-16-01020-f017:**
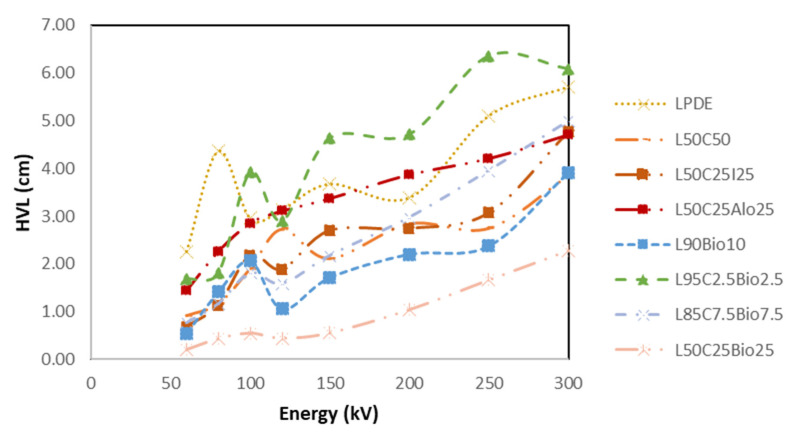
Comparison of HVL of the X-ray properties for all developed composites compared to pure LDPE.

**Figure 18 polymers-16-01020-f018:**
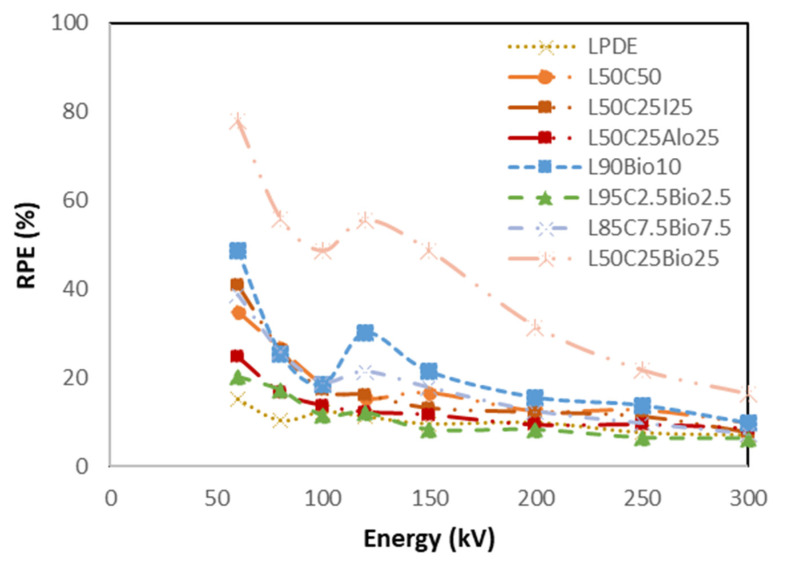
Comparison of RPE of the X-ray properties for all developed composites compared to pure LDPE.

**Table 1 polymers-16-01020-t001:** Formulations of various composites.

No.	Sample Code	LDPE (%)	Cement (%)	Fe_2_O_3_ (%)	Al_2_O_3_ (%)	Bi_2_O_3_ (%)
1	LDPE	100	X	X	X	X
2	L95C5	95	5	X	X	X
3	L85C15	85	15	X	X	X
4	L50C50	50	50	X	X	X
5	L95C2.5I2.5	95	2.5	2.5	X	X
6	L85C7.5I7.5	85	7.5	7.5	X	X
7	L50C25I25	50	25	25	X	X
8	L95C2.5Alo2.5	95	2.5	X	2.5	X
9	L85C7.5Alo7.5	85	7.5	X	7.5	X
10	L50C25Alo25	50	25	X	25	X
11	L90Bio10	90	X	X	X	10
12	L95C2.5Bio2.5	95	2.5	X	X	2.5
13	L85C7.5Bio7.5	85	7.5	X	X	7.5
14	L50C25Bio25	50	25	X	X	25

**Table 2 polymers-16-01020-t002:** X-ray energy range used for the study.

Shortened Name	Tube Potential (kV)	Effective Energy (keV)	Additional Filtration Thickness			
			mm Pb	mm Sn	mm Cu	mm Al
N-60	60	47.9	-	-	0.631	3.912
N-80	80	65.2	-	-	1.980	3.8
N-100	100	83.3	-	-	5.027	3.920
N-120	120	100	-	1.013	5.027	3.950
N-150	150	118	-	2.605	-	3.903
N-200	200	165	1.028	3.004	2.032	3.901
N-250	250	207	3.099	2.062	-	3.925
N-300	300	248	5.152	3.016	-	3.929

**Table 3 polymers-16-01020-t003:** Tensile properties for the most promising results for X-ray shielding.

Sample Code	Tensile Strength (MPa)	Elongation at Break (%)	Tensile Strength Relative to LDPE (%)	Elongation at Break Relative to LDPE (%)
LPDE	15.08 ± 0.59	350.17 ± 11.59	100.00	100.00
L50C50	16.39 ± 0.34	11.34 ± 1.93	108.69	3.24
L50C25I25	9.55 ± 0.18	7.88 ± 1.86	63.33	2.25
L50C25Alo25	13.24 ± 0.27	4.68 ± 0.44	87.80	1.34
L90Bio10	14.15 ± 0.53	332.74 ± 4.94	93.83	95.02
L95C2.5Bio2.5	12.34 ± 0.83	57.45 ± 2.15	81.83	16.41
L85C7.5Bio7.5	13.3 ± 0.3	35.45 ± 4.29	88.20	10.12
L50C25Bio25	13.3 ± 0.17	14.58 ± 1.88	88.20	4.16

**Table 4 polymers-16-01020-t004:** Bismuth composite.

No.	Sample Code	LDPE (%)	Cement (%)	Bi_2_O_3_ (%)
1	L90Bio10	90	X	10
2	L95C2.5Bio2.5	95	2.5	2.5
3	L85C7.5Bio7.5	85	7.5	7.5
4	L50C25Bio25	50	25	25

## Data Availability

Data are contained within the article.

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
