# Peer review of "Novel Polymer Composites for Lead-Free Shielding Applications"

_polymers, 2024, doi:10.3390/polym16071020_

Round 1
Reviewer 1 Report
Comments and Suggestions for Authors
Attached you can find my comments.

Reviewer 2 Report
Comments and Suggestions for Authors
The research work is interesting and contains elements of novelty. Recommended for publication.
It is advisable to remove unnecessary tables and provide references to the literature.
Comments on the Quality of English LanguageIn general, there are no special questions about the quality of the English language in the text
Reviewer 3 Report
Comments and Suggestions for Authors
The paper is about using material shielding characteristics. I think this paper can be published after a minor revision.
1)in the material section, I think it is better for authors to say why this type of material was selected for this test and they highlighted from where they provide them but please don't say that in the text and put a list to it is become attractive for everyone looking for a similar study.
2) Table 1 was repeated two times and only tables should be placed on a page
3) methodology is better described in steps
4) what is this on page 5 "Error! Reference source not found.." and many pages repeated
5) The caption of Fig.2 was placed on another page ! also please
give a reason why for the power less than 80 Kev, this filter is not working
6) you mentioned that :
"As for the MFP which the material thickness to reduce the radiation to 36.8%. "
did you study the thickness of all the material? and how much we should increase the thickness? in addition, what is the negative effect of increasing thickness? finally, I think after a special thickness we cannot see any effect on absorption, maybe my Idea is not true, what did the authors find in their study?
7) don't use solid lines for all diagrams in a Fig, such as Fig. 4 and 5 and other Fig.s
8)" to support further the SEM micrographs of the developed composites. [1717]" Ref 1717!!!!!
Comments on the Quality of English Languagemany typos error and grammar problem can be see
Round 2
Reviewer 1 Report
Comments and Suggestions for Authors
After revisions, I suggest to accept the paper in the present form.